# DeepSketcher: Internalizing Visual Manipulation for Multimodal Reasoning

## Abstract

The "thinking with images" paradigm represents a pivotal shift in the reasoning of Vision Language Models (VLMs), moving from text-dominant chain-of-thought to image-interactive reasoning. By invoking visual tools or generating intermediate visual representations, VLMs can iteratively attend to fine-grained regions, enabling deeper image understanding and more faithful multimodal reasoning. As an emerging paradigm, however, it still leaves substantial room for exploration in data construction accuracy, structural design, and broader application scenarios, which offer rich opportunities for advancing multimodal reasoning. To further advance this line of work, we present DeepSketcher, a comprehensive suite comprising both an image–text interleaved dataset and a self-contained model. The dataset contains 31k chain-of-thought (CoT) reasoning trajectories with diverse tool calls and resulting edited images, covering a wide range of data types and manipulation instructions with high annotation accuracy. Building on this resource, we design a model that performs interleaved image–text reasoning and natively generates "visual thoughts" by operating directly in the visual embedding space, rather than invoking external tools and repeatedly re-encoding generated images. This design enables tool-free and more flexible "thinking with images". Extensive experiments on multimodal reasoning benchmarks demonstrate strong performance, validating both the utility of the dataset and the effectiveness of the model design. The DeepSketcher suite will be released.

## 1 Introduction

Recent progress shows that integrating step-by-step reasoning into VLMs has substantially improved their performance on complex tasks (Meng et al., 2025; Yang et al., 2025a; Xiaomi, 2025; Zhang et al., 2025a; Deng et al., 2025b; Chen et al., 2025). However, current VLMs often exhibit a "thinking over seeing" tendency (Li et al., 2025b): while they can generate lengthy and seemingly coherent reasoning traces, these traces are frequently detached from the actual visual input. In many cases, the models misinterpret critical details in the image or even hallucinate content that is not present (Tu et al., 2025; Sun et al., 2025), suggesting that their reasoning is driven more by linguistic priors than by genuine visual perception (Guan et al., 2024; Fu et al., 2025).

To address this, OpenAI has introduced a new axis for VLM reasoning with "thinking with images" (OpenAI, 2025c). Instead of merely generating textual reasoning traces that overlook visual content, this approach enables models to actively interact with images through an explicit mechanism. By zooming, cropping, and performing systematic image-level manipulations, VLMs are encouraged to ground their reasoning in actual visual evidence. This paradigm represents a shift from "thinking over seeing" to "thinking through seeing," enabling models to analyze visual information more deeply, more thoroughly, and ultimately achieve more reliable multimodal reasoning. Following such an idea, recent efforts have explored stimulating the use of visual information in the reasoning process to enhance model performance in perception and reasoning tasks. VILASR (Wu et al., 2025) defines a closed set of drawing operations and trains the model to decide when to invoke each of them. At inference time, the model selects an operation from this set and predicts the spatial coordinates required to execute it. DeepEyes (Zheng et al., 2025) and OPENTHINKIMG (Su et al., 2025) leverage end-to-end reinforcement learning to incentivize "thinking with images." In this setting, the model learns to actively manipulate visual inputs, such as zooming and cropping; additional related approaches and references are provided in Appendix A. Despite their differences, these ap-

proaches share a common limitation: the supported action space remains relatively restricted, and they inevitably rely on accurate spatial grounding, which remains challenging: curated data seldom yield perfectly accurate annotations, and end-to-end reinforcement learning rollouts are similarly error-prone. To overcome the constraints of a limited action space and to expand the model's "thinking space," another line of work makes a conceptual leap from execution to imagination, aiming to unify generation and reasoning within a single model (Li et al., 2025a;a; Yang et al., 2025d). However, this enlarged thinking space comes at the cost of extremely high training difficulty, and the methods' effectiveness has not yet been thoroughly validated on public benchmarks (Qiao et al., 2024; Xiao et al., 2024; Lu et al., 2023; Wang et al., 2024; Zhang et al., 2024). These methods provide promising directions for visual reasoning in VLMs, while also exposing fundamental trade-offs involving action space, grounding, training feasibility, as well as the inherent difficulty of constructing reliable data for supervision.

To offer a complementary perspective within this paradigm, we introduce the DeepSketcher suite. The first component of the suite is a high-quality dataset with image–text interleaved chain-of-thought trajectories, where textual reasoning steps are interleaved with `<tool_call>` instructions that return visually edited images, serving as auxiliary visual cues to guide subsequent reasoning. A distinctive feature of this dataset is that all images are code-rendered. Specifically, the source images are generated directly from rendering code, while intermediate images are obtained by modifying the source code according to the given instructions and re-rendering the updated code. This code-based approach provides both controllability and semantic clarity, enabling visual manipulations that are precise, reproducible, and less noisy than pixel-level editing, as illustrated by the running example in Figure 1, which contrasts direct code-space editing with grounding-based and generation-based manipulations. Beyond the complex "reasoning → tool call instruction → image manipulation → reasoning" pipeline within our dataset, we further propose a self-contained architecture to internalize the whole thinking mode into a single model. Specifically, the model manipulates images directly within the visual embedding space, allowing seamless integration of visual and textual reasoning. This design eliminates the need for code execution, external tool calls, and repeated re-encoding of images, thereby enabling more flexible "thinking with images" patterns. In summary, the main contributions of this work are as follows:

- By presenting the **DeepSketcher** suite, we provide a complementary perspective within the "thinking with images" paradigm, showing how dataset and model design can jointly support more reliable and flexible multimodal reasoning.

- We construct a high-quality dataset with interleaved image–text chain-of-thought trajectories. All images are code-rendered, and manipulations are conducted in code space, supporting a broad spectrum of open-vocabulary visual operations while avoiding the grounding noise inherent in previous datasets.

- We design a self-contained model that internalizes the "reasoning → tool call → image manipulation → reasoning" chain. This design removes reliance on external tool calls, eliminates the need for coordinate-level predictions, and generalizes beyond code at inference.

## 2 DEEPSKETCHER

### 2.1 THE DEEPSKETCHER DATASET

**Overview of the data curation pipeline.** The DeepSketcher dataset contains extended interleaved image–text reasoning traces, where textual requests (e.g., highlighting a region or adding an auxiliary line) are followed by corresponding visual edits. Each trajectory thus alternates between natural language reasoning and image modifications, encouraging models to ground their reasoning in visual evidence and enabling more thorough multimodal understanding.

Prior approaches to constructing such data typically fall into two categories. (i) Grounding-based manipulation, where models predict an operation target, for example, by outputting a structured action such as $\{'name' : 'image\ zoom\ in\ tool', 'bbox' : [360, 280, 640, 560]\}$ or by generating editing code to perform image modifications (Hu et al., 2024; Zheng et al., 2025). In both cases, the core mechanism relies on accurate prediction of spatial coordinates. (ii) Generation-based manipulation, where image generation models are leveraged to fulfill editing instructions (Chern et al.,

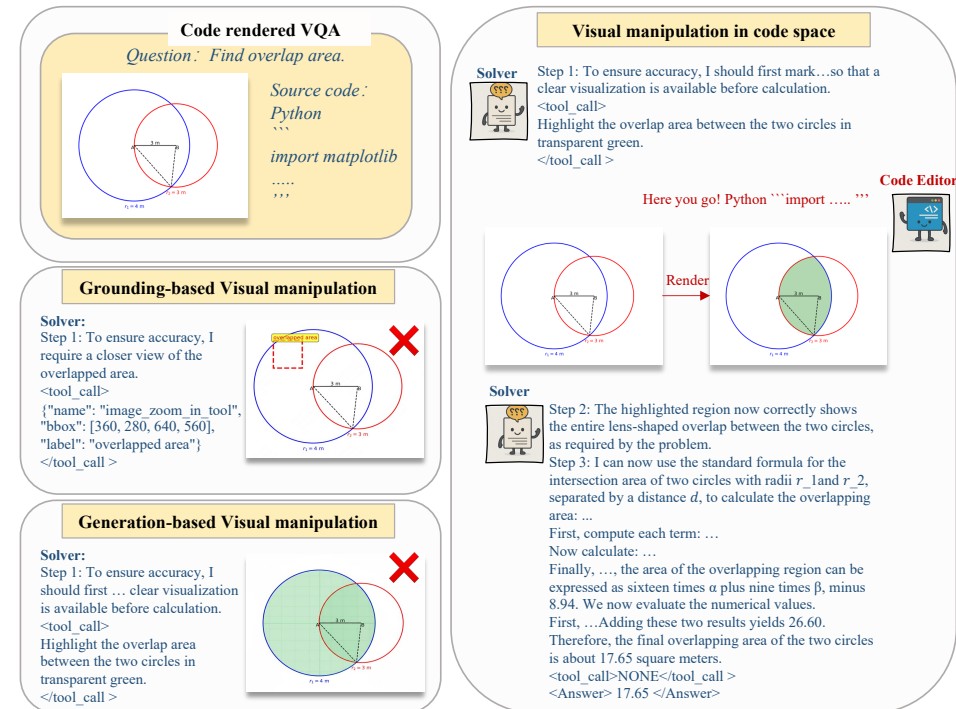

Figure 1: In code space (right), edits are specified through rendering code, offering precision and reproducibility. In contrast, grounding-based manipulation (bounding box predicted by GPT-5 OpenAI (2025b)) and generation-based manipulation (image generated by Nano-Banana Google (2025)) often yield noisy results, underscoring their limitations in stability and controllability.

2025). While effective in some cases, these paradigms often suffer from grounding noise and limited accuracy, making it difficult to obtain consistent, controllable traces as shown in Figure 1.

In contrast, our approach leverages code-rendered VQA data as the foundation. Each instance in our VQA dataset is represented as a tuple $(C, I, Q, A)$, where $C$ denotes the rendering code, $I = \mathcal{R}(C)$ is the rendered image from the renderer $\mathcal{R}$, $Q$ is a visual question, and $A$ is the corresponding answer. This representation is particularly valuable because (i) visual manipulations can be expressed as edits to $C$, ensuring accurate and reproducible modifications, (ii) the alignment between code and image avoids the spatial ambiguity and grounding noise, and (iii) the expressiveness of the code space supports a wide spectrum of operations, rather than limited to a predefined closed set. Building on this representation, we construct the DeepSketcher dataset, which spans diverse domains (Figure 2) and includes a wide variety of actions paired with their corresponding image edits (Table 3).

To curate this dataset, we employ a two-round pipeline. In the first round, we design an automatic agentic system in which two proprietary LLM experts collaborate to solve code-rendered VQA problems. Their dialogues are collected and reformatted into 6k interleaved image–text CoT trajectories, which serve as seed data. We then train an intermediate reasoning model on this seed set to learn proper tool use and to reliably respond to visual manipulation feedback. In the second round, we broaden data diversity by constructing additional code–image pairs from off-the-shelf VQA datasets through code conversion, expanding the dataset to 31k examples. The intermediate model is then deployed within the agentic system, generating richer trajectories and more diverse instruction–edited image pairs without incurring the prohibitive API cost of large-scale agentic interaction. These outputs are subsequently used to train the next-stage model, described in Section 2.2. In the following, we detail the data collection pipeline and the agentic system.

**Data Collection.**    Our data collection follows one fundamental principle: all images must be code-rendered. Although such data may appear difficult to obtain, synthetic visual datasets are already plentiful. Prior work has extensively explored the use of synthetic visual data to enhance VLMs' perception and reasoning abilities, showing its effectiveness across structured domains and beyond (Jia et al., 2025; Wang et al., 2025; Deitke et al., 2024; Cai et al., 2024b; Yang et al., 2025c). This

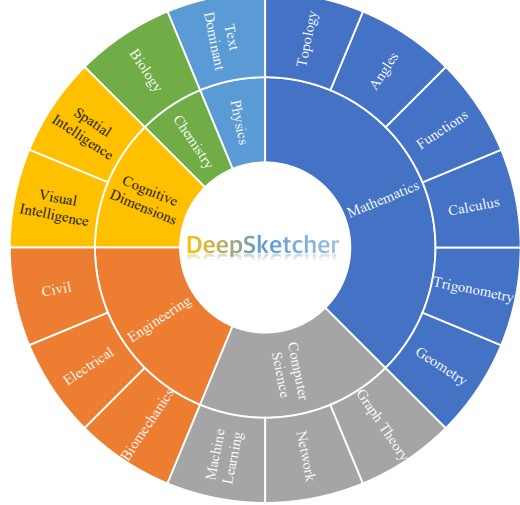

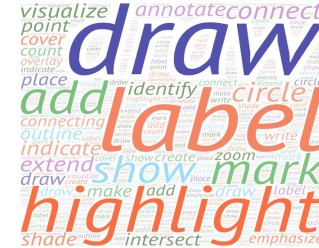

Figure 3: Wordcloud of visual manipulations.

| Rank | Category | Count | Share (%) |
|------|----------|-------|-----------|
| 1 | Labeling/Annotation | 12,340 | 20.9 |
| 2 | Highlighting | 10,437 | 17.7 |
| 3 | Color Operations | 7,383 | 12.5 |
| 4 | Circle Drawing | 6,942 | 11.8 |
| 5 | Line Drawing | 6,919 | 11.7 |
| 6 | Point Marking | 3,924 | 6.6 |
| 7 | Area/Region Operations | 2,641 | 4.5 |
| 8 | Shape Drawing | 2,549 | 4.3 |
| 9 | Others | 5,919 | 4.3 |
| | **Others** | 4,853 | 10 |
| | **Total** | 59,054 | 100.0 |

Figure 2: Disciplinary coverage of our dataset.

Table 1: Distribution of visual manipulations.

---

**Algorithm 1** Agentic curation with *Solver* ($LLM_S$) and *Code Editor* ($LLM_E$)

---

**Require:** Initial code $C_0$, renderer $\mathcal{R}$, question $Q$, max steps $T_{max}$
1: $I_0 \leftarrow \mathcal{R}(C_0)$; $\mathcal{D}_S \leftarrow \{Q\}$; $\mathcal{D}_E \leftarrow \emptyset$
2: **for** $t = 0$ to $T_{max}$ **do**
3:     $(R_t, A, Act_t) \leftarrow LLM_S(\mathcal{D}_S, I_t)$    // CoT $R_t$; $A$ and $Act_t$ are mutually exclusive
4:     **assert** $(A = \varnothing) \oplus (Act_t = \varnothing)$    // enforce exclusivity
5:     Append $(I_t, R_t, Act_t)$ to $\mathcal{D}_S$
6:     **if** $A \neq \varnothing$ **then**
7:         **return** $\mathcal{D}_S$ with $A$
8:     **else**
9:         $C_{t+1} \leftarrow LLM_E(C_t, Act_t, \mathcal{D}_E)$    // complete edited code
10:        Validate $C_{t+1}$ (syntax/render checks); if invalid, repair or backoff
11:        $I_{t+1} \leftarrow \mathcal{R}(C_{t+1})$
12:        Append $(C_t, Act_t)$ to $\mathcal{D}_E$
13:     **end if**
14: **end for**
15: **return** $\mathcal{D}_S$ with $A$

---

makes it a generally applicable strategy for improving visual understanding. In the first round, we sample from CoSyn-400k (Yang et al., 2025b), a large-scale dataset of code–image–QA triples, where all images are code-rendered with associated code and at least one LLM-generated QA. In the second round, we broaden source diversity by converting images from additional VQA datasets (e.g., MMK12 (Meng et al., 2025), UniGeo (Chen et al., 2022), MM-Math (Sun et al., 2024), GeoQA8k (Chen et al., 2021)) into rendering code (`img2code`) and reusing the same pipeline to obtain more varied traces. To ensure the validity and quality of both the CoSyn-400k and `img2code` data, we apply multiple verification and filtering steps, with details given in Appendix B.

**Agentic system for data curation.** After collecting code-image-QA pairs from Coysn-400k, we curate reasoning traces with a two-agent collaborative framework. The system involves two LLM experts with complementary roles: a *Solver* $LLM_S$ that conducts step-by-step visual reasoning, and a *Code Editor* $LLM_E$ that edits rendering code according to natural-language instructions provided by $LLM_S$. Specifically, given a code-rendered image $I_0 = \mathcal{R}(C_0)$ and a question $Q$, $LLM_S$ is prompted to reason explicitly and, whenever visual evidence is uncertain or additional views are needed, to issue a free-form edit request $Act_t$ to $LLM_E$ (e.g., "draw a tangent line", "highlight point A in red"). Upon receiving the request and the current source code $C_t$, $LLM_E$ returns a complete edited program $C_{t+1}$, which is rendered into a new image $I_{t+1} = \mathcal{R}(C_{t+1})$ and fed back to $LLM_S$ for the next round of reasoning. The process continues until the *Solver* produces a final answer $A$ or a termination condition is met (e.g., maximum edit steps).

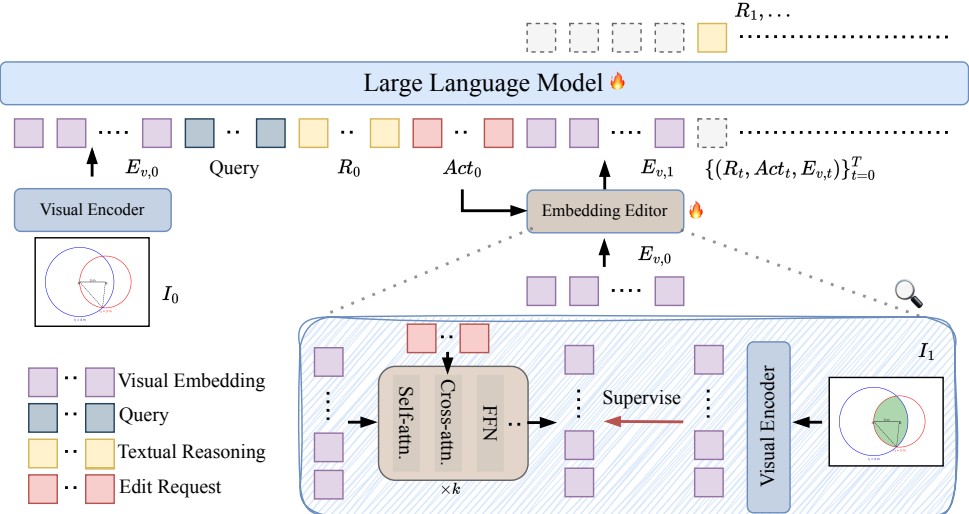

Figure 4: Architecture of the proposed DeepSketcher model. A query $Q$ and initial image $I_0$ are encoded into the vision–language model, producing reasoning tokens $R_t$ and edit instructions $Act_t$. The Embedding Editor manipulates visual embeddings directly, supervised by code-rendered ground-truth edits, and inserts updated embeddings back into the VLM context. This process yields interleaved reasoning and visual manipulation traces, ultimately producing the final answer.

Then, we log the entire interleaved trajectory $\{(I_t, R_t, Act_t)\}_{t=0}^T$, where $R_t$ denotes the *Solver*'s chain-of-thought at step $t$. This yields long image-text CoT traces aligned with code edits and rendered images, which we later standardize into training examples. To improve the reliability of the system, we incorporate mistake-proofing and verification mechanisms; full details are provided in Appendix B. For clarity, the overall procedure is summarized in Algorithm 1.

## 2.2 THE DEEPSKETCHER MODEL

**Overview.** The curated DeepSketcher dataset offers long, interleaved reasoning traces aligned with precise visual manipulations. To leverage this resource, we introduce the DeepSketcher model, specifically designed to integrate such interleaved reasoning with visual operations.

For comparison, common reasoning VLMs take as input an image $I$ and a textual query $Q$, and produce a sequence of textual reasoning steps $\{R_1, R_2, \ldots, R_t\}$ followed by a final answer $A$:

$$\{R_1, R_2, \ldots, R_t\},\ A = \text{LLM}(E_v, Q),$$

where $E_v$ denotes the visual embedding output by the visual encoder and $Q$ denotes the textual query.

In contrast, the DeepSketcher model integrates reasoning and visual manipulation into a unified trajectory. Given an initial image-query pair $(I_0, Q)$, the pair is first encoded by a visual encoder into $(E_{v,0}, Q)$. The model then generates an initial reasoning step $R_0$ (as illustrated in Figure 4). When additional visual clarification is required, it autonomously generates an action $Act_0$. The pair $(E_{v,0}, Act_0)$ is then passed into a built-in *embedding editor*. As shown in the bottom part of Figure 4, it predicts the manipulation directly in the visual embedding space and returns a "manipulated" image representation $E_{v,1}$. The augmented context $\{E_{v,0}, Q, R_0, Act_0, E_{v,1}\}$ is then fed back into the model for subsequent reasoning. This recursive process yields an interleaved trajectory of reasoning, actions, and updated visual embeddings, and finally, the textual answer:

$$(R_0, Act_0, E_{v,1}, R_1, Act_1, E_{v,2}, \ldots, R_{T-1}, Act_{T-1}, E_{v,T}, A) = \text{DeepSketcher}(E_{v,0}, Q).$$

The overview of the pipeline is provided in Figure 4. Next, we detail the training strategy that enables the Deepsketcher model to perform interleaved reasoning and visual manipulations.

**Building the DeepSketcher model.** The training of the DeepSketcher model can be divided into three phases. In the first phase, we directly utilize the features of images in our dataset, rather than

utilizing the visual embedding editor, to warm up the reasoning model. The model is optimized on interleaved image–text sequences. The supervision signal is applied only to textual tokens and `<vision_start>`, `<vision_end>` tokens. These two special tokens serve as boundary markers for visual content in interleaved sequences. Image features are inserted as continuous embeddings and serve only as conditioning context. This enables the model to learn proper structural demarcation between textual and visual modalities during generation.

Formally, consider the $i$-th training example with $T^{(i)}$ images $\{I_0^{(i)}, \ldots, I_{T^{(i)}-1}^{(i)}\}$ interleaved with text. Each image $I_t^{(i)}$ is encoded as a sequence of visual tokens $E_{v,t}^{(i)}$, and we group the intervening text into segments $\{\mathcal{S}_t^{(i)}\}_{t=0}^{T^{(i)}-1}$, where $\mathcal{S}_t^{(i)}$ collects the positions of text tokens that appear after $I_t^{(i)}$ and before $I_{t+1}^{(i)}$. Let $E_{v,\leq t}^{(i)} = \{E_{v,0}^{(i)}, \ldots, E_{v,t}^{(i)}\}$ denote all visual tokens up to image $t$. Each segment is modeled autoregressively, conditioning on the historical text $x_{<\tau}^{(i)}$ and the preceding visual tokens $E_{v,\leq t}^{(i)}$, yielding the per-segment loss $\mathcal{L}_t^{(i)} = -\sum_{\tau \in \mathcal{S}_t^{(i)}} \log P_\theta(x_\tau^{(i)} \mid x_{<\tau}^{(i)}, E_{v,\leq t}^{(i)})$. The phase-1 language modeling objective then averages over all text tokens across the corpus and sums over examples, segments, and token positions:

$$\mathcal{L}_{\text{LM}}^{\text{phase-1}}(\theta) = -\frac{1}{\sum_{i=1}^{N} |\mathcal{S}^{(i)}|} \sum_{i=1}^{N} \sum_{t=0}^{T^{(i)}-1} \sum_{\tau \in \mathcal{S}_t^{(i)}} \log P_\theta\left(x_\tau^{(i)} \,\Big|\, x_{<\tau}^{(i)}, E_{v,\leq t}^{(i)}\right), \tag{1}$$

where $|\mathcal{S}^{(i)}| = \left|\bigcup_{t=0}^{T^{(i)}-1} \mathcal{S}_t^{(i)}\right|$ is the number of text tokens in example $i$. This objective trains the model to issue proper edit requests while ensuring that textual predictions are consistently conditioned on the available visual context.

The editor must handle a broad spectrum of visual modalities (e.g., geometry, charts) and follow diverse instructions to achieve reliable features. Thus, in the second phase, we suggest that larger scale and diverse supervision are indispensable to equip the model with native visual manipulation capabilities. Accordingly, we augment training data constructed via an `img2code` pipeline (detailed in Appendix B.1) to capture the complexity of multimodal reasoning tasks, and we deploy the pretrained reasoning model for the agentic system described in Section 2.2, which yields more training traces enriched with edit-request and image outcome pairs $(Act_t, I_{t+1})$. With these augmentations in place, we finalize the architecture to unify textual reasoning and visual manipulation.

When the model is uncertain about its visual perception, it generates an instruction enclosed by `<tool_call>` tokens. We then extract the hidden states of these tokens, denoted $E_{\text{raw}} \in \mathbb{R}^{N \times D}$, and apply adaptive pooling to obtain a fixed-length sequence $E_{\text{act}} \in \mathbb{R}^{32 \times D}$; the choice of 32 is chosen based on empirical statistics from training data. For the embedding editor, we adopt a Q-Former–style architecture (Li et al., 2023) but drop the text branch and retain an image transformer with cross-attention. Unlike Q-Former, which grounds visual information into a fixed set of learnable query tokens, our module uses visual tokens themselves as queries and injects textual guidance from the action embeddings via cross-attention. Let $E_V \in \mathbb{R}^{K \times D}$ be the visual tokens from the frozen visual encoder. We take queries from $E_V$ and keys/values from $E_{\text{act}}$:

$$Q = E_V W_Q, \qquad K = E_{\text{act}} W_K, \qquad V = E_{\text{act}} W_V,$$

and update the visual tokens via a cross-attention block followed by an FFN:

$$\widetilde{E}_V = \text{MHA}(Q, K, V) + E_V, \qquad E_V^{\text{out}} = \text{FFN}(\widetilde{E}_V) + \widetilde{E}_V.$$

Stacking several such blocks propagates instruction semantics into the visual space, yielding updated visual embeddings $E_V^{\text{pred}} \in \mathbb{R}^{K \times D}$ with the same length $K$ as the input $E_V$.

We perform a second round of training on our proposed model. We initialize from the checkpoint of the reasoning model pretrained in the first stage and freeze all modules except the embedding editor. For supervision, we use the output of the visual encoder on ground-truth edited images as targets, and apply an $\ell_1$ loss to the latent editor's predicted embeddings. Crucially, beginning in this phase, the VLM consumes editor-produced visual tokens rather than ground-truth visual context, and the LM objective is conditioned on the editor's outputs. The phase-2 objective is:

$$\mathcal{L}^{\text{phase-2}}(\theta) = \left\| E_V^{\text{pred}} - E_V^{\text{gt}} \right\|_1 + \mathcal{L}_{\text{LM}}^{\text{phase-2}}(\theta), \tag{2}$$

Table 2: Performance comparison on multimodel reasoning benchmarks.

| Model | MathVerse | Mathvision | MathVista | LogicVista | WeMath | Average |
|---|---|---|---|---|---|---|
| *Proprietary VLMs* | | | | | | |
| Claude3.7-Sonnet (Anthropic, 2025a) | 46.7 | 41.9 | 66.8 | 58.2 | 49.3 | 52.6 |
| GPT-4.1  (OpenAI, 2025a) | 48.9 | 46.4 | 70.4 | 61.1 | 55.5 | 56.5 |
| *Open-source VLMs* | | | | | | |
| InternVL3-8B (Zhu et al., 2025) | 38.5 | 26.3 | 70.4 | 45.6 | 31.7 | 42.5 |
| Qwen2.5-VL-7B (Bai et al., 2025) | 41.1 | 27.0 | 68.2 | 39.8 | 34.3 | 42.1 |
| *Tool-Calling VLMs* | | | | | | |
| VILASR-7B (Wu et al., 2025) | 29.4 | 25.0 | 57.6 | 32.2 | 23.7 | 33.6 |
| DeepEyes-7B (Zheng et al., 2025) | 42.2 | 26.6 | 70.1 | 47.7 | 38.9 | 45.1 |
| *Inner Visual Thought VLMs* | | | | | | |
| Bagel-Zebra-CoT-7B (Li et al., 2025a) | 48.8 | 28.2 | 64.7 | 48.4 | 28.0 | 43.6 |
| Mirage-7B (Yang et al., 2025d) | 27.3 | 28.6 | 63.7 | 40.7 | 16.7 | 35.4 |
| DeepSketcher-7B (Ours) | 43.2 | 32.3 | 69.1 | 48.1 | 37.1 | 46.0 |
| Δ (vs Qwen2.5-VL-7B) | *+2.1* | *+5.3* | *+0.9* | *+8.3* | *+2.8* | *+3.9* |

where $\mathcal{L}_{\text{LM}}^{\text{phase-2}}(\theta)$ is the same as $\mathcal{L}_{\text{LM}}^{\text{phase-1}}(\theta)$ except the visual embeddings.

Compared to the prior approach (Yang et al., 2025d) that edits images in a highly compressed latent space, our method preserves richer semantic information: the editor operates directly on visual tokens with explicit conditioning on action embeddings. This design yields more interpretable guidance and better semantic alignment between textual requests and visual transformations.

In the final phase, we retain the same training objective and unfreeze the LLM backbone to encourage the model to adapt to its own edited outputs, ensuring consistency between generated edit requests and the resulting visual context. The visual encoder is frozen through all three stages.

## 3 EXPERIMENTS

### 3.1 SETUPS

**Baselines.** To evaluate the effectiveness of the proposed DeepSketcher model, we compare it against four categories of baselines: (1) proprietary models, including Claude3.7-Sonnet (Anthropic, 2025a) and GPT-4.1 (OpenAI, 2025a); (2) state-of-the-art open-source models (Zhu et al., 2025; Bai et al., 2025); (3) reasoning VLMs with tool-calling capabilities that rely on external tools (Zheng et al., 2025; Wu et al., 2025); and (4) "thinking-with-generated-images" models that produce inner visual thoughts (Li et al., 2025a; Yang et al., 2025d). Strictly speaking, our model also falls into the fourth category, as it performs interleaved visual-textual reasoning natively, without external tools. We select Qwen2.5-VL-7B as our baseline model.

**Benchmarks.** We evaluate our model on common multimodal reasoning benchmarks, including MathVerse (vision-only) (Zhang et al., 2024), MathVision (mini) (Wang et al., 2024), MathVista (mini) (Lu et al., 2023), LogicVista (Overall) (Xiao et al., 2024), and WeMath (Overall) (Qiao et al., 2024). We also construct an in-house benchmark, Indicator-500, by sampling 500 code-rendered VQA instances from the Cosyn-400k test set. Unlike existing benchmarks, it includes paired code information, which enables to decouple interleaved reasoning from visual manipulation and provides a reliable indicator for the embedding editor during training. (See Appendix C for training details.)

### 3.2 MAIN RESULTS

Table 2 summarizes the performance of different VLMs across the benchmarks described above. For clarity, we group the baselines into four categories as mentioned earlier. The last group, "Inner Visual Thought VLMs," is particularly challenging, as its "thinking space" and "action space" are far larger than those of tool-calling VLMs with fixed utilities. Such models are more sensitive to visual variations, and their robustness in visual manipulation may affect the model performance. When they fail to generate reliable visual content, the resulting noise can propagate through the reasoning trace and hurt overall performance. Despite these challenges, our model consistently outperforms other inner visual thought VLMs such as Bagel-Zebra-CoT-7B and Mirage-7B across most benchmarks. Furthermore, when compared with tool-calling VLMs, despite operating under a substantially more flexible paradigm, it surpasses VILASR-7B and DeepEyes-7B by 12.4 and 0.9

points in average. Together, these results highlight the effectiveness of our approach within this challenging setting, which we attribute to the accuracy and reliability of our training data, and the adaptability of our proposed model architecture.

To better understand the effect of our method, we conduct an in-depth comparison against the base model Qwen2.5-VL-7B. Overall, our approach yields an average improvement of 3.9 points across benchmarks. When breaking down results by task category, consistent patterns emerge: the most reliable gains appear in tasks involving geometry and counting, with particularly striking improvement on MathVision reaching 5.3 points. In addition, math-related problems (LogicVista) involving logical or numerical reasoning also exhibit significant improvements (8.3 points). By contrast, the improvement on tasks that require symbolic manipulation or domain knowledge integration tend to decline, decreasing performance gain to 0.9 points on MathVista. In particular, this dataset contains scientific reasoning and textbook QA, both of which have numerous open-domain images and depend heavily on disciplinary knowledge outside the scope of our training data. Please see the Appendix for more performance details.

## 3.3 ABLATION STUDY

**Ablation study on the agentic data curation system.** For data curation, we design an agentic system where two experts collaborate to solve VQA. The intuition is that, with the aid of a *Code Editor* that has direct access to the source code underlying an image, the solver LLM effectively gains more accurate visual information. With this enhanced context, the solver can tackle problems that would otherwise be unsolvable by independent reasoning, thereby enabling the collection of higher-quality and more informative interleaved reasoning traces. To verify this, we conduct a controlled experiment on a subset of Cosyn-400k under two settings: (i) an *independent answering* setup, where solver LLM works alone, and (ii) a *collaborative answering* setup, where the solver cooperates with another LLM acting as the *Code Editor*. We evaluate the results using the *pass@8* metric, which counts a question as correctly answered if at least one of the eight sampled responses matches the ground truth. As shown in Table 3, both solvers achieve significant gains when paired with the *Code Editor*, confirming that collaboration enables the agentic system to correctly answer more questions, including those that are too challenging for a single model to solve independently. As a result, our data collection pipeline retains more verified examples than the single model. This setup not only increases coverage but also allows us to harvest reasoning traces from more difficult problems that would otherwise be excluded. In this way, the collected trajectories are not only richer in reasoning content but also more challenging and ultimately more valuable for training.

Table 3: Comparison of collaborative vs. independent answers across different LLMs

| Solver | Code Editor | pass@8 |
|---|---|---|
| GPT-4.1 | Null | 0.72 |
| GPT-4.1 | Claude3.7-Sonnet | 0.80 |
| Qwen2.5-VL-72B | Null | 0.67 |
| Qwen2.5-VL-72B | Claude3.7-Sonnet | 0.72 |

**Does the model reason better with the embedding editor?** To evaluate the effect of the embedding editor, we conduct an ablation study across multiple multimodal reasoning benchmarks as well as our in-house Indicator-500 evaluation set (Table 4). We focus on the models obtained in training stage 2 and stage 3, since the stage 1 model does not incorporate the embedding editor. For each model, we consider three experimental settings: (i) deploying the model within an *agentic system* that collaborates with an external *Code Editor* expert, which has direct access to the source code of the input image and thus makes all information explicitly available, this approximates an *upper bound*; (ii) relying solely on the model's built-in embedding editor to manipulate visual representations and generate interleaved reasoning traces; and (iii) bypassing the editor entirely and producing chain-of-thought traces purely in text, which serves as a natural *baseline*, since any degradation relative to this case would imply that the editor introduces noise rather than improving reasoning.

As shown in Table 4, the *embedding editor* consistently improves over the text-only baseline on most benchmarks. A notable exception arises at stage 2: the editor-equipped model underperforms on Indicator-500, trailing the baseline by 4.5 points. This issue is largely alleviated in stage 3, where joint training of the LLM and the editor tightens their coupling. Within stage 3, we see gains under all three settings: the text-only baseline itself is slightly stronger than in stage 2 (41.3 vs 40.6),

Table 4: Ablation study on the embedding editor.

| Stage | Setting | Mathverse | Wemath | Mathvista | Mathvision | LogicVista | Indicator-500 | Average |
|-------|---------|-----------|--------|-----------|------------|------------|---------------|---------|
| 2 | Text-only (*Baseline*) | 37.2 | 28.3 | 65.0 | 28.6 | 45.9 | 38.3 | 40.6 |
|   | Editor | 41.6 | 37.5 | 65.8 | 28.9 | 46.5 | 33.8 | 42.4 |
|   | Agentic (Oracle) | – | – | – | – | – | 41.0 | – |
| 3 | Text-only (*Baseline*) | 38.1 | 31.2 | 65.7 | 33.5 | 41.8 | 37.5 | 41.3 |
|   | Editor | 43.2 | 37.1 | 69.1 | 32.3 | 48.1 | 40.5 | 45.1 |
|   | Agentic (*Oracle*) | – | – | – | – | – | 44.8 | – |

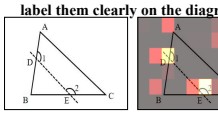
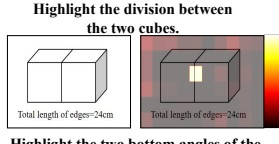

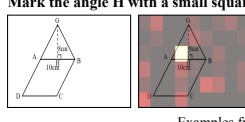
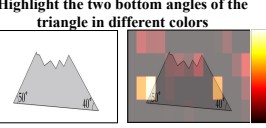

Examples from Indicator-500      Examples from public benchmarks

Figure 5: Difference map visualizations. Each example shows the input image (left), the programmatic rendering (available only in Indicator-500) (middle), and the difference map between the embedding editor output and the original visual embedding (right).

adding the editor lifts the stage 3 average by a further +3.8 (45.1 vs 41.3), and the upper bound is further lifted, reaching 44.8 on *Indicator-500*. Overall, later-stage joint training mitigates the distribution shift from editor-produced visual tokens, yields broad performance gains, and improves the reliability of visual grounding, while the remaining gap to the agentic upper bound underscores opportunities for further optimization and architectural advances.

We are also interested in characterizing how the embedding editor modifies visual content. Since the actual visual outputs are not directly observable, we instead measure the distance between edited outputs and feature embeddings of the input image, and localize pronounced differences as regions of interest (ROIs). We perform this analysis on both our in-house Indicator-500 dataset and the public benchmarks. For Indicator-500, the availability of the underlying source code allows us to issue the exact same instructions to the *Code Editor* expert and obtain programmatically rendered edits for comparison. For the left side of Figure 5, we present results on the in-house Indicator-500 dataset. Because this set provides access to the underlying source code, we can generate programmatic edits to visualize the intended visual modifications. From these results, we observe that the regions of interest generally align well with both the natural language instructions and the programmatic edits, indicating that the embedding editor tends to modify the intended areas. It is worth noting, however, that the programmatic edits in Indicator-500 should be regarded as illustrative rather than absolute ground truth, since natural language instructions may admit multiple valid implementations. On the right side of Figure 5, we show results on the public benchmarks, where no programmatic edits are available. The localized regions of interest still largely align with the intent expressed in the natural language instructions. This observation further reinforces the effectiveness of the proposed module and demonstrates its ability to generalize beyond the code-accessible setting.

# 4 CONCLUSION

We present the DeepSketcher suite as a fresh perspective within the broader paradigm of "thinking with images." At the heart of this suite lies a carefully constructed dataset, where chain-of-thought reasoning is interleaved with code-rendered visual edits—precise, reproducible, and semantically grounded, free from the grounding noise that plagues pixel-level manipulations. Building upon this foundation, we design a self-contained model that internalizes the entire cycle of reasoning, tool invocation, and image manipulation. By removing reliance on external tools and fragile coordinate predictions, the model demonstrates a new pathway toward a resilient multimodal intelligence. Together, these contributions point toward a future where machines learn to "think" with images in a more integrated way.

## 5 ETHICS STATEMENT

We acknowledge that large language models in general may reflect biases present in their pretraining data. In our study, however, all training data are deliberately restricted to mathematics-related tasks and disciplinary benchmarks, drawn exclusively from open-source datasets. Moreover, both the training and evaluation are conducted using open-source frameworks, ensuring transparency and reproducibility.

The design of our model is not intended to introduce concerns related to health, safety, personal security, or privacy: it operates entirely on domain-specific data, avoids the use of personal or sensitive information, and is confined to research-oriented applications. By focusing on well-defined academic tasks, our work contributes to responsible AI research while offering potential benefits for education, scientific discovery, and the broader study of multimodal reasoning.

## 6 REPRODUCIBILITY STATEMENT

We provide the necessary information to facilitate the reproducibility of our results. The main paper describes the data curation process (Section 2.1), the design of the proposed model (Section 2.2), the experimental setups (Section 3.1), and the ablation studies (Section 3.3). Additional details are provided in the appendix, including further information on data curation (Section B), training procedures (Section C), and extended qualitative and quantitative discussions of the experiments (Section D). We commit to publicly releasing the *DeepSketcher* suite (model and dataset) as well as the code for constructing the agentic system to facilitate further research.

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

# A RELATED WORK

## A.1 VISUAL PROMPTING

Visual prompting (VP) is an interaction paradigm that predates the recent surge of multimodal reasoning. Prior studies have demonstrated that incorporating pixel- or region-level cues—such as bounding boxes, markers, scribbles, or segmentation masks—into input images can substantially enhance a model's perceptual capabilities (Yang et al., 2023; Hu et al., 2024; Cai et al., 2024a; Yan et al., 2024), thereby improving its understanding of visual inputs. For instance, SoM (Yang et al., 2023) shows that augmenting input images with labeled cues significantly improves referring and localization performance in GPT-4V. Similarly, Sketchpad-style pipelines automatically compose visual prompts by leveraging a toolbox of detectors and segmenters (with lightweight Python glue) to draw boxes and masks either prior to or during inference, thereby strengthening both perception and downstream reasoning (Hu et al., 2024). However, early VP methods are typically developed on frontier models, since many open-source alternatives lack the capacity to reliably invoke external tools and effectively reason over the resulting feedback. This limitation highlights the importance of equipping models with the ability to internalize such skills, rather than relying solely on externally orchestrated prompting mechanisms.

## A.2 VISION LANGUAGE MODEL REASONING

Enhancing the reasoning ability of vision-language models (VLMs) is a key focus of current VLM research. Following the success of GRPO (Shao et al., 2024; Guo et al., 2025) in textual reasoning, a growing body of work leverages reinforcement learning (RL) to explicitly encourage and strengthen the reasoning skills of VLMs, yielding promising progress (Meng et al., 2025; Deng et al., 2025b; Yang et al., 2025a; Zhang et al., 2025a; Xiaomi, 2025). Nevertheless, most existing approaches remain predominantly oriented toward textual reasoning steps, treating the visual input merely as a static condition rather than an integral component of the reasoning pipeline. To move beyond simply seeing images and to reason more deeply about them, recent studies have introduced a tool-use paradigm, where external vision functions or specialized modules are invoked to manipulate visual inputs—for example, through cropping or zooming—and the edited artifacts are subsequently fed back into the model to guide the next stage of reasoning (Zhang et al., 2025b). This paradigm allows models to better perceive and localize fine-grained image regions, thereby improving visual question answering (VQA) accuracy. However, such "thinking with images" approaches only enable models to perform a restricted set of visual operations, thereby constraining their reasoning space. In parallel, another line of research has sought to expand this reasoning space by incorporating image generative models (Li et al., 2025a; Yang et al., 2025d). Yet, these efforts have largely been confined to limited scenarios such as jigsaw puzzles and mazes, which restrict their broader applicability.

# B MORE DETAILS ON DATASET CONSTRUCTION.

## B.1 IMG-TO-CODE PIPELINE

There is an inherent drawback in the current code-rendered dataset. Specifically, the distribution of questions differs substantially from those authored by human experts in high school, university, or competition settings. Moreover, it is difficult to reliably assess the difficulty of the generated problems, and the perspectives adopted in question construction often lack the nuance and pedagogical intent typically found in human-authored questions. The ability to tackle more challenging problems is precisely why reasoning models are needed.

To bridge this gap, we design an additional `img2code` pipeline in the expansion round to incorporate more realistic and challenging problems. An overview of the pipeline is shown in Figure 6. Concretely, we first sample VQA data from a collection of math-dominant datasets (Meng et al., 2025; Chen et al., 2022; Sun et al., 2024; Chen et al., 2021). We then employ FigCodifier (Wang et al., 2025), a model specifically trained to convert images into code, to process these samples. The resulting code is subsequently rendered back into images.

It is worth noting that `img2code` is an extremely challenging task, as it requires the model to faithfully capture all fine-grained details in an image using programmatic language. We conduct

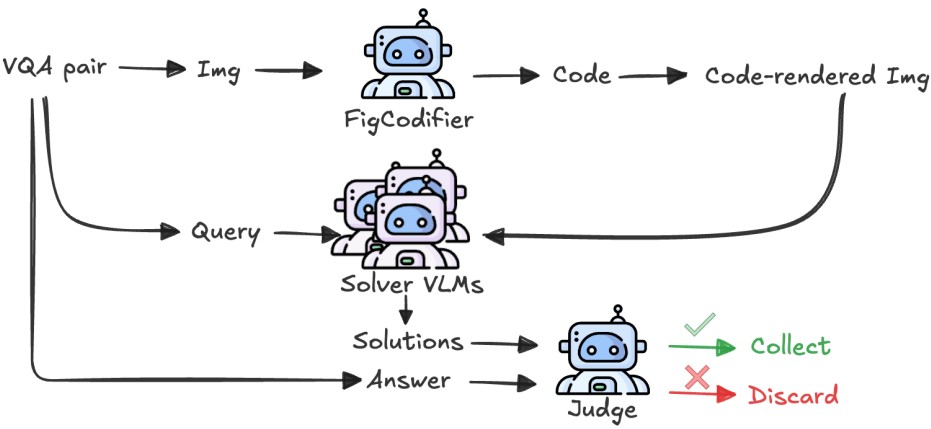

Figure 6: Overview of `img2code` pipeline.

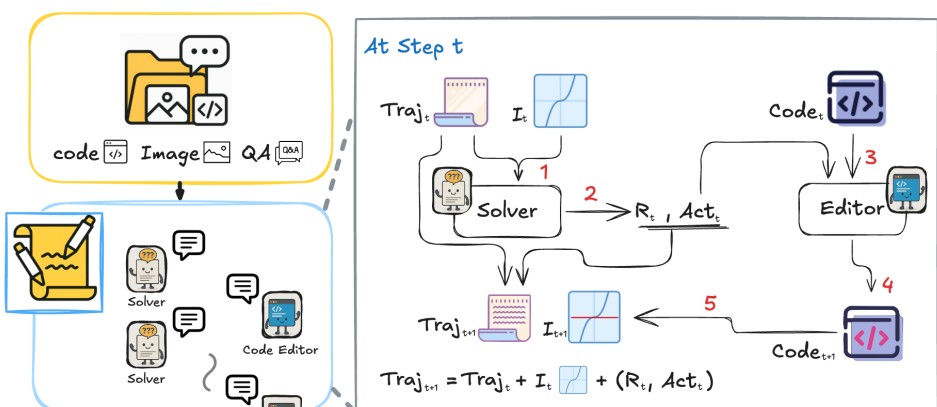

Figure 7: Overview of the DeepSketcher data curation pipeline. We first construct a dataset of VQA problems with images rendered directly from code. An agentic system is then designed to generate interleaved image–text reasoning traces.

preliminary tests using three models: FigCodifier, GPT-4.1, and Claude 4.0-Sonnet (Anthropic, 2025b). By human inspection, the success rates of all three are below 10%. This is because even a minor error in the rendering process can lead to a drastically different semantic meaning of the image. Here, we employ an intuitive compromise for automatic quality filtering. Specifically, we leverage multiple solver VLMs, including GPT-4.1, Claude 3.7-Sonnet, and Qwen2.5-VL-72B, to answer the VQA questions using the re-rendered images. If a solver model can produce a correct answer, then we deem the re-rendered image acceptable: although it may not perfectly replicate the original, it still contains sufficient information for solving the problem.

## B.2 DATA FILTERING

Here, we describe our data filtering process in detail. In the first round of curation, we selected samples from the math, graphic, diagram, and chart subsets of the Cosyn-400k dataset. Although this dataset provides a large volume of diverse data, all samples are synthesized: both questions and answers are generated by LLMs. Consequently, the validity of the provided answers cannot be guaranteed. To mitigate this, we employ several LLM experts, including GPT-4.1, Qwen2.5-72B-VL, and GPT-4.1-mini, to independently answer each question. For every question, we sample two responses from each LLM. If at least one of these responses matches the original answer provided by Cosyn-400k, we retain the question–answer pair; otherwise, we discard it.

For the agentic system, we design several fail-safe loops and verification strategies. First, if the code produced by the *Code Editor* fails to execute during rendering, the erroneous code together with the

error logs are sent back to the editor for another round of editing. By leveraging the error logs, the editor can dynamically adjust its edits, thereby mitigating issues arising from either model mistakes or inconsistencies in the execution environment. If the code is rendered successfully, we then prompt the *Solver* LLM to critically inspect and challenge the rendered content rather than simply accepting it. If the content does not satisfy the *Solver*'s requirements, the *Solver* generates revised instructions for another round of editing.

Finally, before model training, we apply a rejection sampling strategy (Dong et al., 2023). Specifically, we use the base model Qwen2.5-VL-7B to answer all queries and discard those for which it produces the correct answer, retaining only the more challenging cases for training.

### B.3 PROMPT TEMPLATE.

We show the prompt template used in the agentic data curation system in Figure 9, Figure 10, and Figure 11.

## C TRAINING DETAILS

We adopt Qwen2.5-VL-7B (Bai et al., 2025) as the base model. Our implementation is built on LLaMA-Factory (Zheng et al., 2024). The training is carried out in three stages: first, the intermediate tool-calling model is trained on the seed data for 5 epochs with a learning rate of $5 \times 10^{-6}$; next, the embedding editor is trained on the full dataset for 10 epochs with a learning rate of $1 \times 10^{-4}$; finally, the LLM backbone and the embedding editor are jointly trained for an additional 2 epochs with a learning rate of $5 \times 10^{-6}$. The learning objective of the first stage is:

$$\mathcal{L}_{\text{LM}}^{\text{phase-1}}(\theta) = -\frac{1}{\sum_{i=1}^{N} |\mathcal{S}^{(i)}|} \sum_{i=1}^{N} \sum_{t=0}^{T^{(i)}-1} \sum_{\tau \in \mathcal{S}_t^{(i)}} \log P_\theta \left( x_\tau^{(i)} \,\Big|\, x_{<\tau}^{(i)}, E_{v,\leq t}^{(i)} \right), \tag{3}$$

which is mentioned in Section 2.2. The learning objective of the second and third stages is:

$$\mathcal{L}_{\text{LM}}^{\text{phase-2}}(\theta) = -\frac{1}{\sum_{i=1}^{N} |\mathcal{S}^{(i)}|} \sum_{i=1}^{N} \sum_{t=0}^{T^{(i)}-1} \sum_{\tau \in \mathcal{S}_t^{(i)}} \log P_\theta \left( x_\tau^{(i)} \,\Big|\, x_{<\tau}^{(i)}, f_{\text{editor}}\big(E_{v,\leq t}^{(i)}\big) \right). \tag{4}$$

The main difference is that, in stages two and three, the VLM takes as input visual tokens produced by the editor instead of ground-truth visual context, and the LM objective is conditioned on the editor's output.

## D DISCUSSIONS

### D.1 MORE VISUALIZATION RESULTS ON PUBLIC BENCHMARKS

In Section 3.3, we examined "where the model is looking" by visualizing difference maps on our in-house benchmark. Here, we extend this analysis with additional examples on public benchmarks in Figure 8 for a more comprehensive view. As shown in Figure 8a, the difference maps generally align with the model's textual edit intent. A notable case appears in the bottom-right example from MathVista, where the editor correctly attends to the faces of all three individuals in accordance with the instruction, despite the model being trained exclusively on code-rendered images that contain no such open-world scenarios. This result suggests that the model exhibits a certain degree of generalization, as it can attend to novel cases far beyond its training distribution. Then, we turn to failure cases in Figure 8b, where the model's attention seems to deviate from the intent expressed in natural language instructions. These examples might demonstrate the current limitations of the editor and the remaining challenges in faithful visual manipulation.

### D.2 EFFECT OF DECOUPLED MULTI-STAGE TRAINING

To assess the effect of our multi-stage training strategy, we compare checkpoints with and without explicitly decoupling the training of LLM and the embedding editor. As shown in Table 5, the

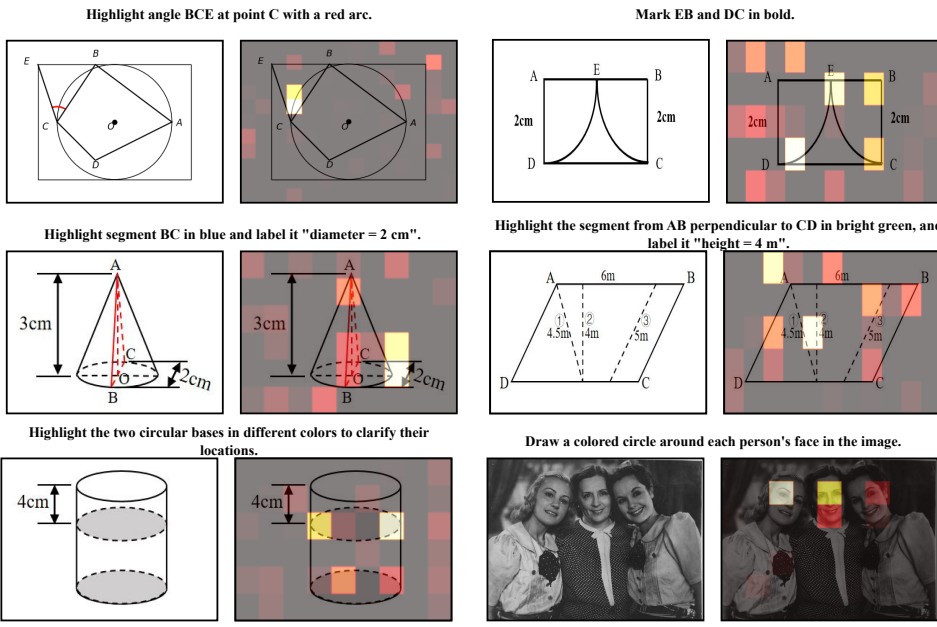

(a) Cases where the model generally attends to the region in accordance with the instruction.

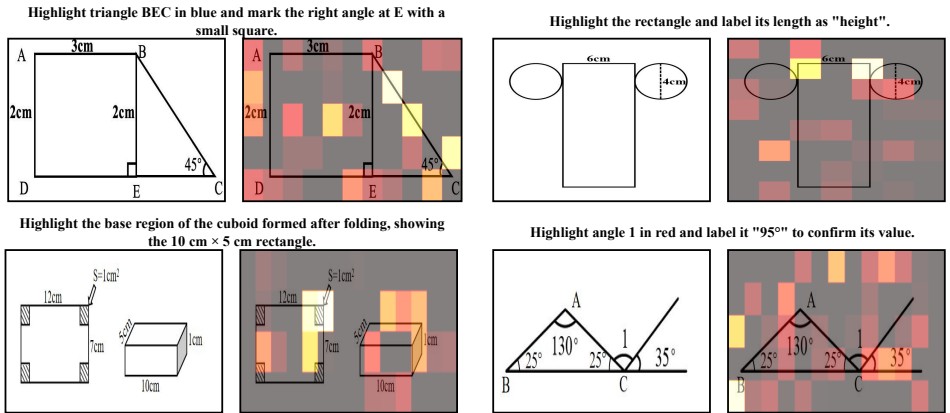

(b) Cases where the model's attention deviates from its textual intent.

Figure 8: Difference map visualization from public benchmarks. (a) Alignment between attention and instruction. (b) Cases with deviation from textual intent.

Table 5: Effect of decoupled multi-Stage training.

| Setting | Mathverse | Wemath | Mathvista | Mathvision | LogicVista | Indicator-500 | Average |
|---|---|---|---|---|---|---|---|
| Single stage training | 39.6 | 36.9 | 66.2 | 25.7 | 45.6 | 39.1 | 42.2 |
| Decoupled training | 43.2 | 37.1 | 69.1 | 32.3 | 48.1 | 40.5 | 45.1 |

three-stage training pipeline—which first pretrains the LLM's reasoning ability, then introduces the embedding editor in a separate adaptation stage, and finally performs joint refinement—consistently outperforms the single-stage alternative. This result suggests that decoupling the LLM from the

editor during training is essential: it allows the base model to acquire robust reasoning skills before being exposed to the more complex task of interleaving reasoning with visual manipulation.

By contrast, directly training the entire system end-to-end in a single stage leads to weaker overall performance, likely because the model must simultaneously learn high-level reasoning and low-level embedding modification, increasing optimization difficulty and reducing stability.

### D.3 ANALYSIS OF MATHVERSE RESULTS

On the MathVerse (Vision-only) benchmark, our model achieves an accuracy of 43.2, outperforming both the baseline Qwen2.5-VL-7B and several tool-calling VLMs. Notably, Bagel-Zebra-CoT-7B attains an exceptionally high score on this benchmark, substantially surpassing our model and ranking near the top of the MathVerse Vision-only leaderboard, comparable to GPT-4.1. This strong performance can be partially explained by the fact that Bagel-Zebra-CoT-7B is post-trained on Bagel-7B (Deng et al., 2025a), whose base model already achieves a notably high score (45–50 according to our implementations) on the MathVerse benchmark. Therefore, the results of Bagel-Zebra-CoT-7B are in part a reflection of the capability of its foundation model. Despite this, our method consistently achieves a 2.4-point improvement in accuracy over five widely used benchmarks, further validating the effectiveness of our approach.

## E  LIMITATIONS AND FUTURE WORK

The proposed DeepSketcher suite provides a complementary perspective to the "thinking with images" paradigm by curating a dataset constructed entirely from code—ensuring accuracy and avoiding the grounding and image-generation noise—and by designing a self-contained model that circumvents reliance on external APIs. Nevertheless, this solution comes with several inherent limitations.

First, the dataset is generated exclusively from code, which may limit the approach's applicability to broader, open-world domains. Moreover, since all questions are automatically generated, there is little fine-grained control over aspects such as difficulty, style, or even the correctness of model-provided answers during reasoning. This lack of precision in data quality raises the risk of "rubbish in, rubbish out," making it crucial to design comprehensive filtering pipelines to ensure the model learns from high-quality content. In this work, we introduced multiple filtering mechanisms and an `img2code` framework to mitigate these effects, but future efforts should focus on expanding data collection to more diverse and open-world domains while improving the quality of generated content.

Second, although our model design removes the dependence on external tools and the need to re-encode images repeatedly, it also diverges from unified generative understanding models in that its "visual thoughts" reside purely in the embedding space. As a result, the actual intermediate visual content remains inaccessible, limiting our ability to fully interpret and analyze the model's behavior. Future work should explore more expressive ways to represent and manipulate visual information, thereby enhancing transparency and interpretability in reasoning within this paradigm.

## F  LLM USAGE

**Use of LLMs in research workflow**   During data curation, LLMs were employed as two collaborating experts to address VQA problems and generate interleaved reasoning trajectories. For model training, we developed a vision–language model (VLM) capable of performing interleaved reasoning based on these curated data. In evaluation, LLMs were further used as judges to assess the correctness of generated answers. All outputs involving LLMs were carefully reviewed and validated by the authors to ensure reliability and accuracy.

**Code implementation**   LLMs' assistance included suggesting solutions to specific programming challenges and providing debugging support. All code produced with LLM assistance was thoroughly reviewed, manually verified, and tested by the authors to ensure correctness, efficiency, and full compliance with the project requirements.

.

---

### Solver LLM starting prompt

You are an agent with broad math knowledge and strong image–text reasoning ability.
You may call an auxiliary tool called the *Renderer* to help you visualize or annotate the image (e.g., draw lines, highlight shapes, add labels).
This tool is invoked using the <tool_call> tag, and its purpose is to make your visual reasoning more accurate.

**Output MUST follow this template exactly:**
<THINK>
Step 1: … reasoning …
Step 2: … reasoning …
</THINK>
<tool_call>...</tool_call>
<ANSWER>...</ANSWER>

⚠️ **IMPORTANT RULES** ⚠️
1. If you are less than 99% confident in your answer, you **MUST** call the Renderer by filling `<ACTION_EXEC>` with a specific drawing instruction (e.g., "Draw a red circle around triangle ABC").

2. In that case, <ANSWER> must be exactly **"TBD"**. Do NOT attempt to answer yet.

3. If you are 99% confident in your answer, set `<tool_call>` to **"NONE"** and fill <ANSWER> with the final answer.

4. <tool_call> must only contain **visual drawing instructions** — do NOT include textual, logical, or general suggestions.

5.Any output that breaks these rules will be rejected by the grader.

---

Figure 9: The *Solver* LLM starting prompt.

**Writing assistance** LLMs were utilized to support the preparation and refinement of this manuscript. Their assistance covered tasks such as proofreading for grammatical accuracy, improving sentence flow and clarity, and rephrasing passages to enhance readability. All generated text was carefully reviewed, assessed, and revised by the authors to ensure the accuracy, consistency, and integrity of the final manuscript. The authors retain full responsibility for all statements, interpretations, and conclusions presented in this paper.

## Solver LLM mid prompt

You are continuing the same problem.

The image shown below has been edited according to your previous <tool_call> instruction.

1. First, carefully check whether the visual edits match what you asked for.

    If they do: proceed with the next step of reasoning.

    If they do **NOT** match: adjust your drawing request in <tool_call> to correct it.

2. Do **NOT** repeat earlier reasoning. Resume from the next step number.

    Use *"Step k:"* where $k = last\_step + 1$.

3. Use this exact format:

<THINK>

Step k: …

Step k+1: …

</THINK>

<tool_call>

… (new drawing instruction if still <99% confident, else write NONE) …

</tool_call>

<ANSWER>

… (write the final answer if sure, or TBD if not) …

</ANSWER>

⚠ **RULES** ⚠

1. If you are now ≥99% confident, set <tool_call> to **NONE** and provide the final <ANSWER>.

2. Otherwise, revise or re-use your drawing request in <tool_call> and leave <ANSWER> as **TBD**.

3. Never repeat earlier steps. Always continue from the last step.

Figure 10: The prompt template when *Solver* LLM receives updated visual information.

**Code editor LLM system prompt**

You are CodeEditor-GPT, a strict code-rewriting agent.

Your job: update the entire source file so that it satisfies the natural-language instruction.

★ RESPONSE FORMAT (no extra text!) ★

"```python\n"
  "# (full revised code here)\n"
  "```\n"

⚠ **IMPORTANT RULES** ⚠

1. Do NOT output anything outside the python fenced block.

2. Keep the programming language identical to CURRENT_CODE.

3. Output the **entire updated file**; you may copy unchanged lines verbatim, but add, delete, or reorder anything needed to satisfy the instruction.

4. If the request is impossible, reply exactly: CODE_ERROR.

Here's an example of how to answer the question:

CURRENT_CODE:

```python

import matplotlib.pyplot as plt

plt.figure();          # line-1

plt.show()             # line-2

```python

INSTRUCTION: Add a title "Demo" to the plot.

OUTPUT:

```python

import matplotlib.pyplot as plt

plt.figure()

plt.title("Demo")

plt.show()
```

Figure 11: The prompt template for *Code Editor* LLM.

