# OpenReview forum: "DeepSketcher: Internalizing Visual Manipulation for Multimodal Reasoning"
_ICLR.cc/2026/Conference — ICLR 2026 Conference Withdrawn Submission_

### Official Review · Reviewer_4gNQ · 2025-10-29

**Soundness:** 2
**Presentation:** 2
**Contribution:** 2
**Rating:** 2
**Confidence:** 4

**Summary:**

This paper introduces DeepSketcher, a new suite for multimodal mathematical reasoning. It has two primary contributions: 1) A novel dataset of 31k interleaved image-text Chain-of-Thought (CoT) trajectories, where visual manipulations are generated by editing and re-rendering the source code of an image. 2) A model DeepSketcher-7B with novel architecture featuring an "Embedding Editor" that internalizes this process, allowing the model to perform "visual thoughts" by manipulating visual embeddings directly, rather than relying on external tools.

**Strengths:**

1. Constructing an interleaved CoT dataset from code rendering is a novel and valuable approach. This method enables the generation of fine-grained, decomposable, and high-precision data, which is a significant challenge in the field.
2. The proposed method, editing visual tokens via embedding editor is novel and interesting idea. This can be applicable to other tasks.

**Weaknesses:**

1. **Model Comparison Scope:** The experimental comparison in Table 2 provides valuable context, but could be strengthened by including more directly comparable baselines. While models like VILASR and Mirage are included, their focus on general spatial reasoning (e.g., mazes) differs from DeepSketcher's mathematical reasoning objective. Including direct comparisons to state-of-the-art multimodal math reasoning models, such as Mulberry, TVC, and especially MathCoder-VL, would offer a clearer assessment of DeepSketcher's relative performance. MathCoder-VL seems particularly relevant given its use of a code-rendered dataset methodology and strong reported results.
2. **Lack of Validation and Rigorous Ablation:** I think the paper's core claims about the model and dataset should be further validated.
* **In-house Benchmark:** The indicator-500 dataset is used for evaluations and ablations, but there is no mention of it being human-validated to reduce noise and ensure quality.
* **Model Ablation Clarification:** The ablation study for the "Embedding Editor" (Table 4) aims to show its benefit. However, the "Text-only (Baseline)" might not fully isolate the editor's contribution from the textual reasoning learned from the dataset. An alternative baseline, such as SFT-ing the base model (Qwen2.5-VL) solely on the text-only CoT traces from the DeepSketcher dataset, could provide a more direct comparison and potentially offer stronger evidence for the editor's specific advantages.
* **Dataset Contribution Validation:** Demonstrating the value of the DeepSketcher dataset independently of the specific DeepSketcher model would be beneficial. The pass@8 metric for data curation offers some insight, but may be a limited proxy for overall dataset utility. Providing results from training other standard models (e.g., (1) base VLMs like Qwen2.5-VL with text-only SFT, (2) tool-calling models like PixelReasoner or Ground-R1 [1, 2], (3) intrinsic reasoning models like v1 [3] or Bagel-Zebra-CoT-7B) on the DeepSketcher dataset could further solidify its contribution to the field.
3. **Readability:** Some sections of the paper could benefit from clearer structuring. For instance, dividing the content into more subsections might help improve readability and the logical flow of the arguments.

---
References:
1. Pixel Reasoner: Incentivizing Pixel-Space Reasoning with Curiosity-Driven Reinforcement Learning, Su et al., NeurIPS 2025
2. Ground-R1: Incentivizing Grounded Visual Reasoning via Reinforcement Learning, Cao et al., arXiv 2025
3. v1: Learning to Point Visual Tokens for Multimodal Grounded Reasoning, Chung et al., arXiv 2025

**Questions:**

1. Given that the DeepSketcher dataset are generated via LLM agents, what steps were taken to assess the quality and correctness of the data beyond automated checks (like pass@8)? Could the authors conduct or report on a human evaluation study for both the main DeepSketcher dataset and the indicator-500 subset to quantify potential noise or errors?

---

### Official Review · Reviewer_7dcu · 2025-10-31

**Soundness:** 3
**Presentation:** 2
**Contribution:** 2
**Rating:** 4
**Confidence:** 5

**Summary:**

This paper introduces DeepSketcher, a suite that includes a dataset and a model for multimodal reasoning under the "thinking with images" paradigm. The dataset has 31k image-text interleaved reasoning trajectories, with code-rendered data. The model manipulates visual embeddings directly to generate visual thoughts. Experiments on benchmarks like MathVerse and LogicVista show it outperforms most open-source VLMs and inner visual thought models, with an average 3.9-point improvement over the base model Qwen2.5-VL-7B. It does especially well on geometry and logical reasoning tasks, proving both the dataset’s utility and the model’s effectiveness.

**Strengths:**

1. The idea of the paper is easy to understand and effective.

2. The paper is well-written and easy to follow.

3. The proposed data generation pipeline is clear and easy to scale up

**Weaknesses:**

1. My key question is: why do we need to do latent reasoning in this setting? As the author mentioned in the data generation pipeline, they successfully constructed a pipeline to map thinking and action into code manipulation of the given images. What motivates the authors to use an embedding editor in their training rather than simply using this pipeline to map it back and encode the edited image again? Training such an editor is extremely costly.

2. The improvement of the proposed method is not that sound. The authors mentioned some baseline models, but training with different data, I do not think they can be used as a fair comparison. The author should include SFT/RL baselines with same amount of data, and tool calling baseline using the correct tool. For example, given the tool calling prompt of the code manipulation, how will qwen2.5-vl perform in a zero-shot manner?

3. Performance comparison with basic CoT is needed. Also, the author mentions several type of image manipulation data, I would like to know how the model solves problem with these tools, including the distribution of the type of tools they use and whether they succeed with these tools. For example, drawing correct line. These will help validate the true contribution of the proposed framework.

**Questions:**

As stated in weakness part.

---

### Official Review · Reviewer_JyuM · 2025-11-01

**Soundness:** 3
**Presentation:** 3
**Contribution:** 3
**Rating:** 6
**Confidence:** 3

**Summary:**

This paper presents DeepSketcher, providing a 31K training dataset and a self-contained model for thinking with images on math problems. The dataset contains 31k chain-of-thought (CoT) reasoning trajectories with diverse tool calls and resulting edited images, covering a wide range of data types and manipulation instructions with high annotation accuracy. Building on this resource, they design a model that performs interleaved image–text reasoning and natively generates ''visual thoughts'' by operating directly in the visual embedding space, rather than invoking external tools and repeatedly re-encoding generated images.

**Strengths:**

Overall It's a good paper, for providing a useful training dataset and showing it working on math problems. It's quite interesting to see that thinking-with-images training could improve math performance. The experiment is also illustrative, with Table4 comparing text only training data v.s. multimodal training data.

**Weaknesses:**

More baselines should be added. E.g. Gemini 2.5 Pro, GPT5, GPT o3, Visual SketchPad, ...
Could also consider evaluate on IsoBench, to evaluate how much improvement on text wise and how much on image wise.

**Questions:**

See weaknesses.

---

### Official Review · Reviewer_VysT · 2025-11-02

**Soundness:** 3
**Presentation:** 2
**Contribution:** 2
**Rating:** 4
**Confidence:** 4

**Summary:**

This paper introduces DeepSketcher, a comprehensive suite for multimodal reasoning that advances the "thinking with images" paradigm through two key contributions: a dataset of 31k interleaved image-text chain-of-thought trajectories with code-rendered visual manipulations, and a self-contained model that performs visual reasoning by directly manipulating representations in the embedding space. The authors show the effectiveness of the approach through extensive experiments.

**Strengths:**

- The code-based data curation approach effectively addresses grounding noise and enables diverse visual operations through code-space manipulations.
- The embedding editor internalizes visual manipulation in the embedding space, eliminating the need for external tools and repeated image re-encoding.
- The approach achieves consistent improvements across benchmarks with thorough ablations and visualizations validating proper visual grounding.

**Weaknesses:**

- The proposed framework is quite limited in addressing the VQA tasks where the input image can be code-rendered, such as math reasoning.
- The paper’s novelty is somehow incremental, and the authors didn’t discuss several highly relevant works [1,2]. The authors should also include the related work section in the main paper.
- While the work has compared the effectiveness of embedding editor, the analysis is still shallow and insufficient. For example, the paper did not extensively discuss the code-redering and image manipulation accuracy for the fine-tuned model, and how these nuanced factors affect the final performance.
- In addition, it would be beneficial if the authors could provide an in-depth analysis of the remaining errors of the proposed framework.
- minor: The citation format in the caption of Figure 1 seems to be wrong.

[1] ReFocus: Visual Editing as a Chain of Thought for Structured Image Understanding. ICML 2025.

[2] OctoTools: An Agentic Framework with Extensible Tools for Complex Reasoning. 2025.

**Questions:**

See the weaknesses section above.

---

### Note · Authors · 2025-11-12

I have read and agree with the venue's withdrawal policy on behalf of myself and my co-authors.